# Intra- and Intertumoral Microglia/Macrophage Infiltration and Their Associated Molecular Signature Is Highly Variable in Canine Oligodendroglioma: A Preliminary Evaluation

**DOI:** 10.3390/vetsci10060403

**Published:** 2023-06-19

**Authors:** Ryan G. Toedebusch, Ning-Wei Wei, Kulani T. Simafranca, Jennie A. Furth-Jacobus, Ingrid Brust-Mascher, Susan L. Stewart, Peter J. Dickinson, Kevin D. Woolard, Chai-Fei Li, Karen M. Vernau, Frederick J. Meyers, Christine M. Toedebusch

**Affiliations:** 1Department of Surgical and Radiological Sciences, School of Veterinary Medicine, University of California, Davis, CA 95616, USA; rgtoed@ucdavis.edu (R.G.T.); nwwei@ucdvais.edu (N.-W.W.); ksimafranca@ucdavis.edu (K.T.S.); jfurthjacobus@ucdavis.edu (J.A.F.-J.); pjdickinson@ucdavis.edu (P.J.D.); cfli@ucdavis.edu (C.-F.L.); kmvernau@ucdavis.edu (K.M.V.); 2Department of Anatomy, Physiology and Cell Biology, School of Veterinary Medicine, University of California, Davis, CA 95616, USA; ibrustmascher@ucdavis.edu; 3Division of Biostatistics, School of Medicine, University of California, Davis, CA 95616, USA; slstewart@ucdavis.edu; 4UC Davis Comprehensive Cancer Center, Sacramento, CA 95817, USA; fjmeyers@ucdavis.edu; 5Department of Pathology, Microbiology, and Immunology, School of Veterinary Medicine, University of California, Davis, CA 95616, USA; kdwoolard@ucdavis.edu; 6Department of Internal Medicine, Division of Hematology and Oncology, Center for Precision Medicine, Microbiology, and Immunology, School of Medicine, University of California, Sacramento, CA 95817, USA

**Keywords:** dog, one health, galectin-3, transforming growth factor beta-1, glioblastoma, pediatric high-grade glioma

## Abstract

**Simple Summary:**

Canine oligodendrogliomas are universally fatal primary brain tumors. Glioma-associated microglia/macrophages (GAMs) have been shown to contribute to immunosuppression and tumor progression in human glioblastoma (GBM). While a robust GAM infiltrate has been observed in canine oligodendrogliomas, their corresponding molecular signature has not previously been explored. The results of this study show that GAMs variably infiltrate canine oligodendrogliomas. We observed marked differences in GAM density within individual tumors and across tumors. We further observed elevations in several GAM-derived pro-tumorigenic molecules, suggesting that GAMs likely contribute to canine oligodendroglioma pathogenesis. However, similar to GAM density, the tumor tissue expression of the majority of molecules assayed demonstrated significant variability. This is in contrast to our previous work on canine astrocytomas, which more consistently demonstrated robust increases in several GAM-derived pro-tumorigenic molecules. This study raises the possibility that the immune microenvironment across oligodendrogliomas and astrocytomas has key differences which may be relevant to future therapeutic targeting.

**Abstract:**

The goal of this study was to define the glioma-associated microglia/macrophage (GAM) response and associated molecular landscape in canine oligodendrogliomas. Here, we quantified the intratumoral GAM density of low- and high-grade oligodendrogliomas compared to that of a normal brain, as well as the intratumoral concentration of several known GAM-derived pro-tumorigenic molecules in high-grade oligodendrogliomas compared to that in a normal brain. Our analysis demonstrated marked intra- and intertumoral heterogeneity of GAM infiltration. Correspondingly, we observed significant variability in the intratumoral concentrations of several GAM-associated molecules, unlike what we previously observed in high-grade astrocytomas. However, high-grade oligodendroglioma tumor homogenates (n = 6) exhibited an increase in the pro-tumorigenic molecules hepatocyte growth factor receptor (HGFR) and vascular endothelial growth factor (VEGF), as we observed in high-grade astrocytomas. Moreover, neoplastic oligodendrocytes displayed robust expression of GAL-3, a chimeric galectin implicated in driving immunosuppression in human glioblastoma. While this work identifies shared putative therapeutic targets across canine glioma subtypes (HGFR, GAL-3), it highlights several key differences in the immune landscape. Therefore, a continued effort to develop a comprehensive understanding of the immune microenvironment within each subtype is necessary to inform therapeutic strategies going forward.

## 1. Introduction

Canine oligodendrogliomas are universally fatal primary brain tumors. With a disease prevalence of 0.9% for all glial neoplasms [1], oligodendrogliomas account for approximately 23% of all canine primary brain tumors [1]. Brachycephalic breeds including boxers, Boston terriers, bullmastiffs, and bulldogs are overrepresented, comprising nearly 60% of reported oligodendrogliomas [2]. Even with surgical resection and radiation therapy, the median survival time remains dreadfully short at 9–14 months [3,4,5]. Treatment resistance is likely multifactorial, as it is in human high-grade gliomas. For example, canine glioma tumor mutational rate is exceedingly low [6], providing protection from anti-tumor immune responses. Moreover, glioma-associated microglia/macrophages (GAMs) have been shown to contribute to immunosuppression and tumor progression in adult glioblastoma (GBM) [7]. A robust GAM infiltrate has been observed in canine oligodendrogliomas [8,9], but their corresponding molecular signature remains unknown.

In human GBM, microglia and macrophages are recruited to the tumor by glioma-derived chemoattractants (e.g., chemokine ligand-2 (CCL2) [10,11]). As the most abundant infiltrating immune cell population, GAMs comprise approximately 30% of the total tumor mass, up to as much as 50% in some cases [12]. Moreover, under the influence of glioma cells, GAMs become a major source of molecules (e.g., interleukin-1 beta (IL-1β) [13], interleukin-10 (IL-10) [7], vascular endothelial growth factor (VEGF) [14]) that support tumor growth and contribute to an immunosuppressive tumor microenvironment. We previously demonstrated that high-grade canine astrocytomas exhibit a similar GAM response to human GBM. Increased GAM density correlates with tumor grade and a marked increase in GAM chemoattractant CCL2 [15]. Moreover, we observed a pro-tumorigenic molecular signature rich in GAM-associated molecules interleukin-10 (IL-10), vascular endothelial growth factor (VEGF), and hepatocyte growth factor receptor (HGFR) [15].

While the glioma–GAM signaling axis is complex, transforming growth factor-beta 1 (TGFβ1) has been identified as a key signaling molecule mediating tumor invasion, migration, and immunosuppression in GBM [16,17]. Furthermore, this abundant cytokine polarizes infiltrating GAMs to a tumor-supportive phenotype that promotes tumor malignancy. In particular, microglia-derived TGFβ1 stimulated glioma cell invasion into surrounding normal brain parenchyma in rodent models [18] and human glioma [19]. It is likely that microglia-derived TGFβ1 also contributes to canine glioma cell malignancy, as our laboratory observed robust TGFβ1 immunoreactivity (IR) at the leading edge of the tumor mass corresponding with increased GAM density in canine astrocytoma [15]. Moreover, TGFβ1 increased canine glioma cell migration and invasion in vitro [15]. Therapeutic targeting of TGFβ1 has demonstrated efficacy in rodent models of GBM but has failed to translate into effective human patient therapies. However, given its prominent pro-tumorigenic role in GBM, there are ongoing efforts to refine the therapeutic targeting of TGFβ signaling in GBM [20] which may also be applied to canine high-grade glioma.

As tissue context and the tumor microenvironment apply selection pressures which influence tumorigenesis and tumor progression, further dissecting the similarities and differences between canine astrocytomas and oligodendrogliomas may highlight the need for divergent therapeutic strategies across glioma subtypes. Therefore, the goals of this study were to (1) quantify intratumoral GAM density through immunofluorescence and (2) quantify the intratumoral concentration of several known GAM-derived pro-tumorigenic molecules through canine-specific protein arrays and Western blot in canine oligodendrogliomas. 

## 2. Materials and Methods

### 2.1. Sample Collection

Tissue samples from canine oligodendroglioma patients were obtained from clinical cases presented to the Veterinary Medical Teaching Hospital, University of California, Davis, between 2002 and 2018. Case information is provided in Table 1. Samples were collected via surgical or image-guided biopsy or necropsy within 1 h following euthanasia. Normal canine brain tissue was collected at necropsy within 1 h following euthanasia from client-owned dogs donated for research with informed consent. Tissue samples were snap-frozen in liquid nitrogen or immersion fixed in 10% neutral-buffered formalin and paraffin embedded. All tumors were histologically classified by a board-certified veterinary pathologist according to the National Cancer Institute-led multidisciplinary Comparative Brain Tumor Consortium [21] and further subdivided according to the 2021 WHO classification of tumors of the central nervous system [22].

### 2.2. Image Acquisition and Analysis

#### 2.2.1. Immunofluorescence and Confocal Microscopy

Formalin-fixed, paraffin-embedded tissue was sectioned at 10 μm and mounted on poly-L-lysine-treated microscope slides. The tissue was deparaffinized using xylene and decreasing concentrations of ethanol. Antigen retrieval was performed using Dako Antigen Retrieval Solution (DAKO) at 95 °C for 20 min. Tissue was permeabilized with Tris-buffered saline with 0.1% Triton X-100 (TBST) and blocked overnight at 4 °C in 5% normal goat serum and 1% bovine serum albumin in TBST. After blocking, tissue was incubated in primary antibody solution, or blocking buffer alone for negative controls, for 24 h at 4 °C. Primary antibodies utilized included: microglia/macrophages: rabbit polyclonal anti-ionized calcium-binding adaptor molecule (IBA-1) (Wako Pure Chemical Industries, Ltd., Chuo-Ku, Osaka, Japan; 019-19741, 1:1000); oligodendroglioma cells: rabbit polyclonal anti-oligodendrocyte transcription factor 2 (OLIG2) (Millipore Sigma, Burlington, MA, USA; AB9610, 1:200); galectin-3 (GAL-3): mouse monoclonal anti-GAL-3 (Abcam, Boston, MA, USA; ab31707, 1:1000). Tissue was then washed three times with TBS with 0.5% Triton X-100 and incubated in secondary antibody solution (IgG heavy and light goat anti-rabbit antibody conjugated with Alexa Fluor 488 (Molecular Probes, Invitrogen, Carlsbad, CA, USA; 1:1000) for one hour at room temperature. Tissue was mounted with Vectashield with 4′,5-diamidino-2phenylindole (DAPI) (Vector Labs, Burlingame, CA, USA). The entire tumor volume, or similar tissue volume in nontumor samples, was imaged on a Leica TCS SP8 STED 3x confocal microscope. Microglial quantification in normal brains was performed on the fronto-parietal lobes of the cerebral cortex.

#### 2.2.2. Microglia Quantification

Microglial quantification was conducted in a double-masked manner. Images were collected as described above (N.-W.W.). Ten to fifteen images per acquisition were selected by a random number generator and given code names. Coded images were dispersed to two trained evaluators (K.T.S. and J.A.F.-J.). GAMs were identified by positive IR to ionized calcium-binding adapter molecule 1 (IBA-1). To ensure cells were not counted twice, the nucleus had to be identified to count the cell. Both counters evaluated all images for each case. Counts were given to a third party for analysis, followed by unmasking of the image identity and stratification into appropriate groups (C.M.T.).

### 2.3. Canine Protein Arrays

Commercially available protein arrays with pre-selected, canine-validated targets were utilized for an unbiased evaluation of the molecular milieu in high-grade canine oligodendrogliomas compared to in a normal canine cortex. Canine tissue was prepared as previously described [23] and analyzed by RayBiotech Life (Peachtree Corners, GA, USA) with standard quality control. In brief, Quantibody^®^ Canine Cytokine Arrays (QAC-CYT-1, QAC-CYT-2, QAC-CYT-4) utilized two non-overlapping arrays of antibody pairs to quantify selected molecules. RayBiotech confirmed no cross-reactivity between antibody pairs and standard controls. See Appendix A for target protein list.

### 2.4. Western Blot Analysis

Protein extracts from normal brain and tumor samples were obtained via homogenization of 30 mg tissue on ice in 300 μL RIPA buffer containing 1X Halt Protease Inhibitor Cocktail (Thermo Fisher Scientific, Waltham, MA, USA) and 1X Halt Phosphatase Inhibitor Cocktail (Thermo Fisher Scientific, Waltham, MA, USA). Protein concentration was determined using BCA assay (Thermo Fisher Scientific, Waltham, MA, USA). Proteins (60 μg) were separated on 10% SDS polyacrylamide gels and transferred to nitrocellulose membranes. Membranes were washed three times in TBS with 1% Tween20 (TBS-Tween20) at room temperature and blocked in blocking buffer (5% fat-free milk diluted in TBS-Tween20) for 1 h at room temperature. Membranes were then incubated in the following primary antibody solutions diluted in fresh blocking buffer at 4 °C overnight: TGFβ1, 13 kD and 44 kD (Abcam, Cambridge, MA, USA; ab92486, 1:2500), TGFβR1 (Santa Cruz Biotechnology, Inc., Dallas, TX, USA; sc-101574, 1:200), TGFβR2 (Santa Cruz Biotechnology, Inc., Dallas, TX, USA; sc-17791, 1:100), SMAD2/3 (BD Biosciences, San Jose, CA, USA; 610842, 1:1000), phospho-SMAD2/3 (Millipore, Burlington, MA, USA; 04-953, 1:500), and GAL-3 (Abcam, Boston, MA, USA; ab31707, 1:1000). Membranes were then washed three times in TBS-Tween20 and incubated with species-specific, HRP-linked secondary antibodies (Jackson Immuno Research Labs Inc., West Grove, PA, USA; 1:20,000) in 1% BSA at RT for 2 h. Chemiluminescence images were acquired using ChemiDoc XRS+ System (Bio-Rad, Hercules, CA, USA) after ECL reagent incubation (Thermo Fisher Scientific, Waltham, MA, USA). Images were analyzed using ImageLab software (Bio-Rad, Hercules, CA, USA). Data are expressed as relative optical densities normalized to control dog tissue.

### 2.5. Statistical Analysis

Statistical analysis was performed with GraphPad Prism v. 8.4.3 and R v. 3.6.3 software. Data were tested for normality via Shapiro–Wilks test. Normally distributed data are presented as the mean ± SEM. An unpaired, two-tailed Student’s *t*-test or ANOVA with Tukey’s multiple comparisons test was conducted for comparisons. The protein array and Western blot data were not normally distributed; data are presented as median and interquartile range. Comparisons were made using the exact Wilcoxon test; tests were performed at the 0.05 level (two sided) with no adjustment for multiple comparisons.

## 3. Results

### 3.1. Microglia and Macrophages Heterogeneously Infiltrate Canine Oligodendrogliomas

To evaluate the infiltration pattern of glioma-associated microglia and macrophages (GAMs) in canine oligodendroglioma, we analyzed confocal microscopy images of paraffin-embedded canine oligodendrogliomas (low grade, n = 5; high grade, n = 5) and normal canine cortex (n = 4) (see Appendix A for patient details). Contrary to the diffuse, homogeneous GAM infiltrate we previously observed in canine astrocytomas [15], intratumoral GAM infiltrate was highly heterogenous (Figure 1). GAM heterogeneity was most pronounced in high-grade oligodendrogliomas (HGGs). Within a single tumor, we observed several areas of sparse IBA-1+ IR (Figure 1C), as well as several areas of dense, robust IBA-1+ IR (Figure 1D).

Estimation plots demonstrated little difference in mean microglia/macrophage counts between counters for normal brain (0.68; 95% CI (−10.1–8.6)), low-grade glioma (0.91; 95% CI (−9.1–10.3)), or HGG (7.7; 95% CI (−30.2–20.5)) (Figure 2A–C). There was no difference in absolute counts between counters for normal brain (*p* = 0.8370), low-grade glioma (LGG) (*p* = 0.8059), or HGG (*p* = 0.4045) (Figure 2D–F). Moreover, there was a positive and strong correlation between each counter’s values for LGG (r = 0.9500, *p* = 0.0067) and HGG (r = 0.9919, *p* = 0.0041) (Figure 2D–F).

Absolute GAM quantification, as determined by the number of IBA-1+ cells per image, was not different between normal brain (27.0 ± 4.0), LGG (36.2 ± 6.9), and HGG (93.3 ± 36.2; *p* = 0.0887) (Figure 3A). The relative GAM density, expressed as the percentage of IBA-1+ nuclei of total nuclei per image, was not different between normal brain (20.8 ± 1.7), LGG (25.6 ± 4.3), and HGG (21.5 ± 7.1; *p* = 0.7409) (Figure 3B). While the group means in absolute GAM quantification were not different, the variance between groups was significantly different (*p* = 0.0019), reflecting the intertumor heterogeneity of GAM infiltration within histopathological grade in canine oligodendroglioma. For instance, the range of GAM counts across tumors within LGG was 39 cells (Figure 3C). This range was further increased in HGG at 151 cells (Figure 3D). The intratumoral GAM heterogeneity within each histopathological grade can also be observed in Figure 3C,D.

### 3.2. The Molecular Landscape of Canine High-Grade Oligodendrogliomas Exhibits Increased Expression of Proteins Associated with a Pro-Tumorigenic GAM Signature

The chemokine ligand 2 (CCL2), also referred to as monocyte chemoattractant protein 1, is a potent inducer of chemotaxis and proliferation of microglia [24] and other immune cells in the glioma microenvironment in human GBM [25]. We did not observe differences in CCL2 protein levels between normal canine brain (n = 3) and high-grade canine oligodendroglioma (n = 6) samples (*p* = 0.381) (Figure 4A). Similarly, no difference in protein expression of granulocyte-macrophage colony-stimulating factor (GM-CSF) was observed between groups (*p* = 0.393) (Figure 4A). However, the variance in GM-CSF levels between groups was significantly different (*p* < 0.0001).

Additionally, we assayed several molecules associated with pro-tumorigenic GAM function in human GBM. Of the growth factors assayed, vascular endothelial growth factor (VEGF) was significantly increased in HGG (n = 6) (9.5; 4–307.7 pg/mL) compared to in normal brain (n = 3) (0; 0–60 pg/mL; *p* = 0.036) (Figure 4B). Hepatocyte growth factor receptor (HGFR), expressed by microglia [26] and glioma cells [27], is a negative prognostic indicator in human GBM. We observed a striking increase in HGFR in HGG (n = 6) (223.5; 7.5–15,040 pg/mL) compared to in normal brain (n = 3) (0; 0–60 pg/mL; *p* = 0.048) (Figure 4B). Cytokine evaluation revealed increased expression of interleukin-12β (IL-12β) in HGG (n = 6) (192.9; 4.1–716.9 pg/mL) compared to in normal brain (n = 3) (0; 0–0 pg/mL; *p* = 0.024) (Figure 4C). 

### 3.3. TGFβ and Galectin-3 May Contribute to Canine Oligodendroglioma Pathogenesis

To determine the influence of TGFβ signaling in canine oligodendroglioma, we assessed the relative protein expression of key proteins in the canonical TGFβ signaling cascade by immunoblotting. Relative protein expression was not different between normal brain (n = 3) and HGG samples (n = 4) across our selected targets (Figure 5A,B, Appendix A). While the median expression of TGFβ receptor I (TGFβRI) was 23-fold higher than in normal brain (*p* = 0.2608), there was a significant difference in variance between groups (*p* = 0.0022) and a marked range of TGFβRI relative protein expression across oligodendroglioma samples (range: 0.1–65 relative optical density). Similarly, the median expression of phosphorylated SMAD 2/3 (pSMAD2/3) was 18-fold higher than in normal brain (*p* = 0.3371), with significant difference in variance between groups (*p* = 0.0011) and a marked range of pSMAD2/3 relative protein expression across oligodendroglioma samples (range: 0.1–58 relative optical density).

Given the crosstalk between TGFβ signaling and galectin-3 (GAL-3) [28,29], we additionally evaluated the relative expression of GAL-3. While the median relative expression of GAL-3 was 32-fold higher in HGG tissue (n = 4) compared to in normal brain (n = 3) (*p* = 0.0608; Figure 5A,C), there was a marked range in GAL-3 expression levels across oligodendroglioma samples (range: 12–56 relative optical density). As noted with many other targets, there was a significant difference in variance between groups (*p* = 0.0010). Immunostaining of HGG for GAL-3 revealed multifocal areas of positive IR throughout the tumors (Figure 5D). All positive IR was adjacent to a nucleus, indicating cellular localization. We did not observe any positive IR for GAL-3 in normal brain tissue (data not shown). There were several OLIG-2-positive cells with co-localization of robust GAL-3 IR (Figure 5D). On the contrary, tumor areas rich in IBA-1 IR had weak GAL-3 IR. Moreover, we did not observe any co-localization with IBA-1-positive cells (Figure 5D).

## 4. Discussion

In this study, we demonstrated that microglia/macrophage infiltration is highly variable within and across tumors in canine oligodendroglioma. Accordingly, the protein expression of key molecules associated with the GAM molecular signature was highly variable in this study cohort. However, individual tumors exhibited markedly increased levels of several growth factors (e.g., VEGF, HGF/HGFR) and pro-tumorigenic cytokines (e.g., IL-10, IL-12β). Additionally, canine HGG exhibited robust expression of GAL-3, as protein expression was several-fold higher in each of the four tumor homogenates evaluated. Moreover, this study suggests that GAL-3 expression predominantly originates from neoplastic oligodendrocytes, not microglia/macrophages.

The most striking finding in this study was the marked intra- and intertumoral differences in GAM density. As shown in Figure 3, we observed dramatic ranges in GAM counts within individual LGG and HGG tumors. Moreover, there was a significant difference in the variance between LGG and HGG GAM density compared to in normal brain. Using similar quantification techniques, we did not observe this degree in variability in canine astrocytomas [15]. Unlike the diffuse, relatively homogeneous infiltration observed in astrocytomas, GAMs tended to have discrete clusters within oligodendrogliomas. Similar to our present findings, previous studies also reported a range in IBA-1+ cellular density in canine oligodendrogliomas [8,9] without a significant difference in IBA-1+ density across tumor grades [8,9].

Another key outcome of this study was the observed difference in the GAM-associated molecular landscape between canine oligodendrogliomas and astrocytomas. We observed significant variability in the intratumoral protein concentration of putative pro-tumorigenic cytokines and chemokines across targets assayed. Moreover, several molecules in the canonical TGFβ signaling pathway had very low expression levels. These findings may be a direct result of the variability in GAM density across tumors, although we were unable to determine this due to limitations in tissue availability, as different cases were utilized for immunofluorescence and protein quantification experiments. It is also possible that the variability in protein concentration was due to sampling bias, as a single tumor homogenate was used for protein evaluation across both studies. However, it is worth considering that there are key differences in the GAM response between canine oligodendrogliomas and astrocytomas.

Despite a relatively quiet cytokine and chemokine response in canine oligodendroglioma, we did observe robust increases in molecules that were also increased in canine astrocytoma [15], including receptor tyrosine kinase pathway molecules VEGF and HGFR. Increased VEGF expression in canine glioma subtypes has been documented in several studies [15,30,31,32], including a recent study demonstrating increased VEGF concentration in the cerebrospinal fluid of glioma-bearing dogs [33]. Notably, increased VEGF is also a shared feature between human GBM [19] and pediatric high-grade gliomas (pHGG) [34]. Additionally, we identified increased protein concentration of HGFR across canine glioma subtypes, although HGFR protein concentrations were much more robustly increased in canine high-grade astrocytomas (mean value: 6172 pg/mL [15]) relative to the HGG in this study (median value: 223 pg/mL). Increased mRNA levels of *MET*, the gene encoding HGFR, has been described in both canine astrocytomas and oligodendrogliomas. However, in contrast to our findings, HGG exhibited the largest increase in *MET* [32]. HGFR has been implicated in mediating a host of pro-tumorigenic functions in human GBM, including stemness [35] and invasion [36]. Importantly, HGFR has been identified as a substrate for the metalloprotease ADAM metallopeptidase 8 (ADAM 8) in temozolomide-treated GBM cells, leading to temozolomide resistance in GBM [36]. While less is known about HGFR in pHGG, approximately 6–10% of tumors analyzed harbor genetic alterations in the *MET* gene, leading to impaired cell cycle regulation [37,38]. Targeted inhibition of HGFR, while promising in preclinical models, failed in both pediatric [38] and adult patients [39,40,41]. However, novel combination therapeutic approaches which include HGFR inhibition may yield more promising results that benefit canine and human patients alike.

A surprising finding in this study was the expression pattern of GAL-3, a chimera-type galectin with myriad functions [42]. We previously observed a marked increase in GAL-3 expression in canine astrocytomas [15]. However, there was marked variability in protein expression across HGG samples in this study. As GAL-3 has been implicated in the alternative activation of macrophages [43] and microglia in human stroke [44], we hypothesized that GAMs would be a major source of GAL-3 in canine oligodendroglioma, similar to in GBM [45]. Contrary to our hypothesis, there was a paucity of GAL-3 IR in regions rich with GAMs, but there was robust IR for GAL-3 co-localizing with neoplastic oligodendrocytes. Therefore, neoplastic oligodendrocytes are the likely source of GAL-3 in canine oligodendroglioma, just as neoplastic astrocytes are the primary source of GAL-3 in pediatric pilocytic astrocytomas [46]. However, GAL-3 is likely to affect GAM function in canine oligodendroglioma, as it is implicated in immunosuppression and resistance to immunotherapies in GBM [47,48]. Importantly, therapeutic targeting of GAL-3 in a rodent model of GBM demonstrated reduced tumor size and the density of pro-tumorigenic GAMs, leading to increased overall survival [49]. Immune checkpoint molecules were concurrently up-regulated on CD8^+^ T cells, suggesting a synergistic effect with an immune checkpoint blockade. Thus, GAL-3 may be a promising therapeutic target shared across canine glioma subtypes and pediatric and adult human glioma. 

As canine glioma is increasingly recognized as a relevant model for adult human GBM, further characterization of the similarities and differences between oligodendrogliomas and astrocytomas has important translational implications. A recent comprehensive molecular analysis of more than 80 tumor samples demonstrated that, as a collective group, canine gliomas are genetically and epigenetically more like pHGG, not GBM. While GBM and pHGG are equally aggressive, they are biologically distinct tumors. GBM, the most common adult primary brain tumor, describes grade IV astrocytic tumors without mutations in genes encoding isocitrate dehydrogenase 1, 2 [22]. Pediatric HGGs are a heterogenous group of tumors that commonly harbor mutations in genes encoding histone variants H3.1 or H3.3 [22], which are not found in GBM. Importantly, GBM and pHGG display divergent immune responses. As the most abundant glioma-invading cells in GBM [50], GAM infiltration is positively correlated with glioma grade [12], invasiveness [51], and resistance to therapy [52]. Under the influence of GBM, GAMs are a major source of inflammatory cytokines (e.g., IL-1β [53], IL-8 [54], IL-10 [55]) and chemokines (e.g., CCL2 [25], CCL4 [56], CCL5 [57]) that support GBM growth [58] and mitigate anti-tumor immune function [10]. Unlike GBM, pHGG GAMs express a paucity of inflammatory cytokines and chemokines [34]. While these GAMs display some evidence of activation, they exhibit markedly reduced levels of inflammation-associated genes compared to GBM (e.g., *CCL5*, *IL1A*, *IL1B*) [34]. Comparison of the GAM response across canine astrocytomas and oligodendrogliomas has raised the possibility that canine astrocytomas more closely align with GBM given the robust increase in several GAM-associated chemokines and cytokines [15], while canine oligodendrogliomas may more closely align with pHGG. Additional studies, including single-cell RNA sequencing and spatial transcriptomics/proteomics, are necessary to further characterize the similarities and differences in GAM function between canine glioma subtypes to inform appropriate translational application of canine glioma clinical trials and appropriate therapeutic approaches for canine glioma patients.

This study had limitations which should be considered upon data interpretation. First, our control population was relatively small. Second, we were not able to age match all the control dogs, particularly for the immunofluorescence studies. The mean age of dogs used for the immunofluorescence studies was 6 ± 1.2 years, contrasting with the mean age of 9 ± 1.1 and 9 ± 1.0 for dogs with LGG and HGG, respectively. Dog age likely has a significant impact on microglial density and phenotype, as we previously reported in canine lumbar spinal cord [59]. Despite limited tissue availability, the relative GAM density observed in this study’s control population was similar to what has been described in rat [60] and canine brain [15] but slightly higher than what has been described in human brain [61]. A third limitation of this study was the lack of matched cases for analysis of microglial quantification and intratumoral concentration of GAM-derived pro-tumorigenic molecules. It would have been ideal to compare the microglial density and distribution to the molecular signature, but we were limited by tissue availability. These limitations provide a rationale for the establishment of a shared tissue biorepository across veterinary institutions. This resource would expand the number of cases included in veterinary studies, adding power to study conclusions, and may accelerate discovery in our field.

## 5. Conclusions

These data highlight the potential similarities and differences in the GAM response between canine oligodendrogliomas and astrocytomas. While there are key differences in these two tumor types, we also identified shared putative therapeutic targets across canine glioma subtypes. Going forward, canine astrocytomas and oligodendrogliomas should be evaluated as distinct tumor types for characterization and target validation studies in order to conduct effective therapeutic trials, as well as appropriately translate canine clinical trial outcomes to human patients. 

## Figures and Tables

**Figure 1 vetsci-10-00403-f001:**
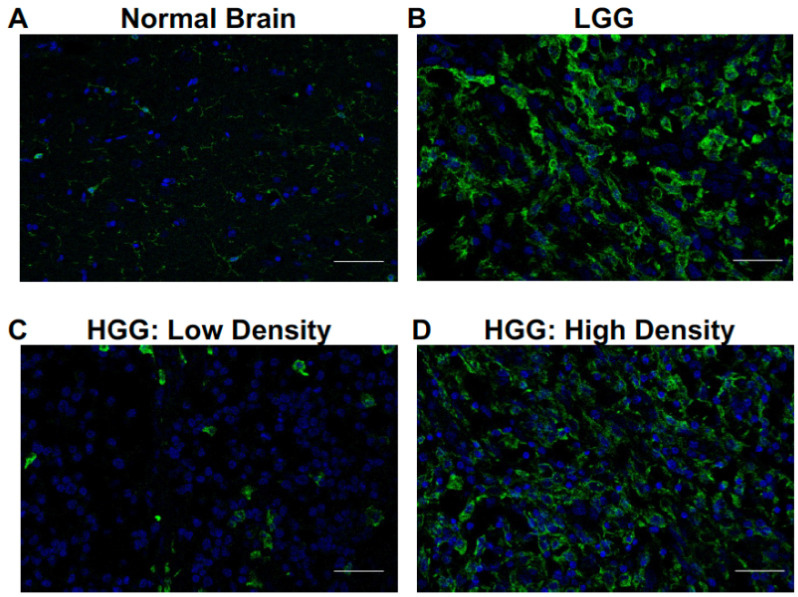
Microglia and macrophages heterogeneously infiltrate canine oligodendrogliomas. Representative immunofluorescence staining of normal brain (n = 4), LGG (n = 5), and HGG (n = 5). (**A**) Microglia, identified by positive IR to ionized calcium-binding adapter molecule 1 (green; IBA-1), were observed among scattered cellular nuclei (blue; DAPI) in normal brain. (**B**) Infiltration of cells with IBA-1 IR consistent with microglia/macrophages was noted in (**B**) LGG and (**C**,**D**) HGG. Most IBA-1+ cells in low-grade oligodendroglioma had thickened, elongated processes compared to IBA-1+ cells observed in normal brain. IBA-1 IR was quite heterogeneous, particularly in high-grade oligodendrogliomas. We observed areas of (**C**) low-density IBA-1+ cells with an amoeboid morphology and (**D**) high-density IBA-1+ cells with both elongated and amoeboid morphology. Scale bars, 20 μm.

**Figure 2 vetsci-10-00403-f002:**
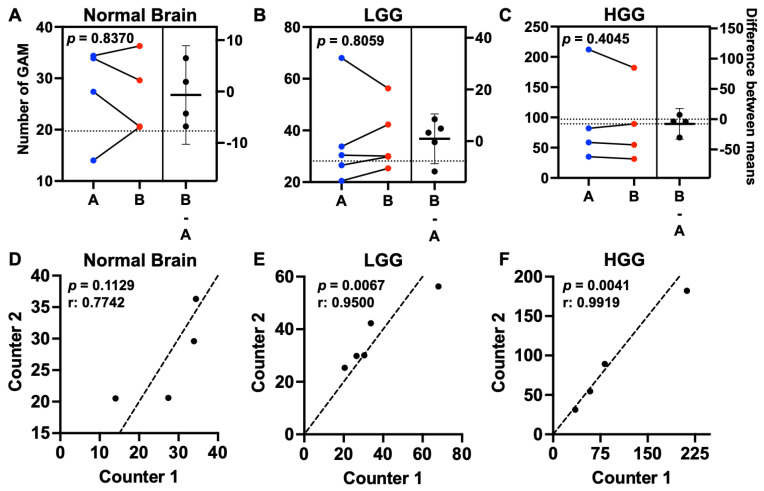
GAM counts were reproducible across counters. Estimation plots showing the mean GAM count per dog for counter A (blue dots) and counter B (red dots) in the left panels for (**A**) normal brain (n = 4), (**B**) LGG (n = 5), and (**C**) HGG (n = 4). There was no difference between mean counter values across groups evaluated. The difference between means is represented by the horizontal line with 95% CI in the right panels. Comparisons based on paired *t*-test. Correlation plots for (**D**) normal brain (r = 0.7742), (**E**) LGG (r = 0.9500), and (**F**) HGG (r = 0.9919). The counter’s values were positively correlated for both LGG (*p* = 0.0067) and HGG (*p* = 0.0041). Correlation coefficient based on paired *t*-test. Black dots represent the mean GAM count per dog for Counter 1 (x- axis) and Counter 2 (y-axis); dashed line represents the line of identity.

**Figure 3 vetsci-10-00403-f003:**
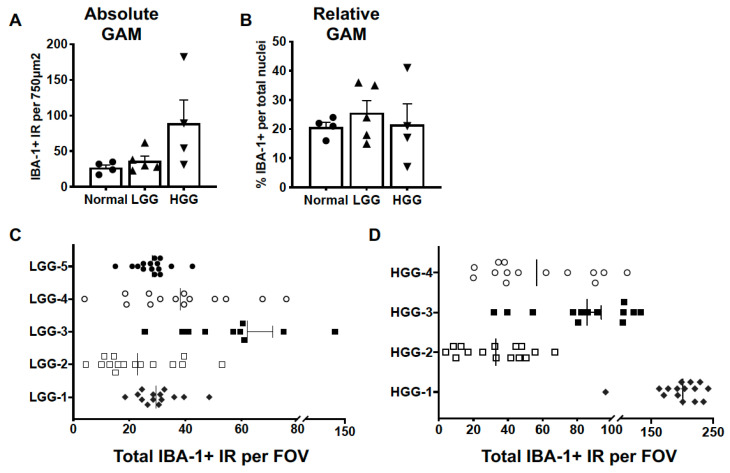
GAM density displayed marked intratumoral and intertumoral variability. We did not observe differences in (**A**) absolute (*p* = 0.0853) or (**B**) relative (*p* = 0.3091) GAM density in either LGG (n = 5) or HGG (n = 4) oligodendrogliomas compared to normal brain (n = 3). Comparisons based on one-way ANOVA with post hoc Tukey’s multiple comparisons test; bars represent group mean with standard error of the mean. Scatter dot plot of the number of counted IBA-1+ cells per field of view (FOV) for each of 5 samples of (**C**) LGG and (**D**) HGG evaluated. The vertical line within each data set represents the median value.

**Figure 4 vetsci-10-00403-f004:**
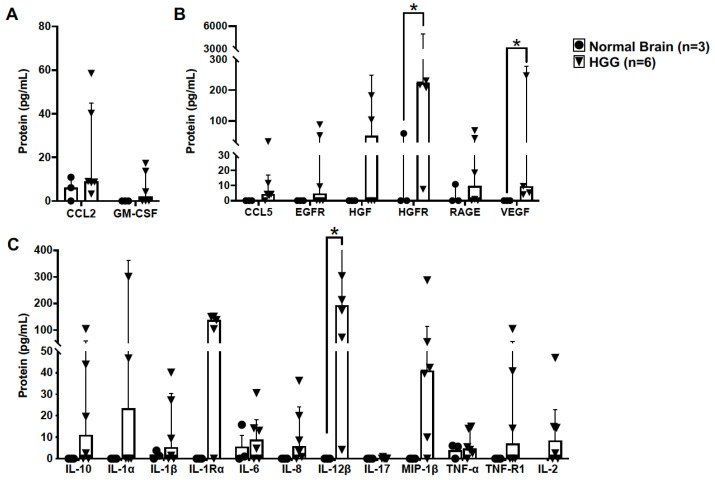
Canine high-grade oligodendrogliomas have increased expression of proteins associated with a pro-tumorigenic GAM signature. (**A**) Protein levels of CCL2 and GM-CSF, key mediators of GAM proliferation and recruitment, were not different in HGG tumor homogenate (n = 6) compared to in normal brain (n = 3) (CCL2, *p* = 0.381; GM-CSF, *p* = 0.393). (**B**) Protein levels of several GAM-derived growth factors were not detectable, or were present in low concentrations, in normal canine brain (n = 3). In contrast, we observed markedly increased levels of hepatocyte growth factor receptor (HGFR; *p* = 0.048) and vascular endothelial growth factor (VEGF; *p* = 0.036) in HGG homogenate (n = 6). (**C**) Similar to growth factors assayed, several cytokines were not detectable, or were present in low concentrations, in normal canine brain (n = 3). However, interleukin-12β (IL-12β) was markedly elevated in HGG homogenate (n = 6; *p* = 0.024). Comparisons based on the exact Wilcoxon test; * *p* < 0.05. Bars represent group median with the interquartile range.

**Figure 5 vetsci-10-00403-f005:**
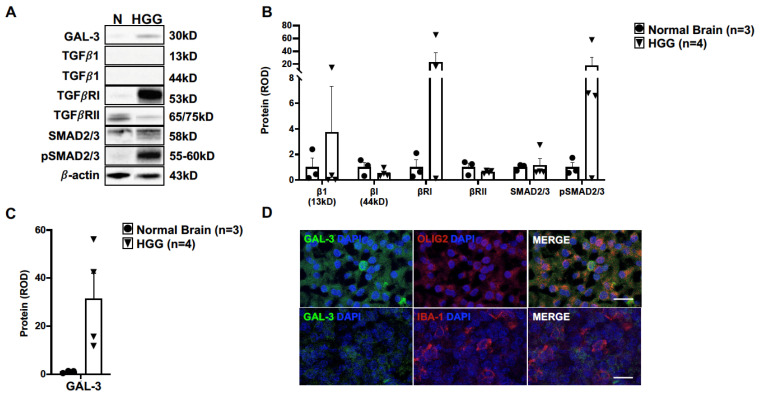
TGFβ and galectin-3 signaling may be relevant in canine oligodendroglioma. (**A**) Representative immunoblots for proteins within the canonical transforming growth factor β (TGFβ) pathway and galectin-3 (GAL-3) in normal canine brain (n = 3) and HGG homogenates (n = 4). Relative protein expression of (**B**) TGFβ targets or (**C**) GAL-3 (*p* = 0.0608) was not different between normal brain (n = 3) and HGG homogenate (n = 4). Comparisons based on unpaired *t*-test; bars represent group mean with standard error of the mean. (**D**) Representative immunofluorescence staining for GAL-3 in HGG. Multifocal areas of robust cytoplasmic GAL-3 IR were observed to co-localize with OLIG2 IR (top panel). Serial sections of tumor areas rich in IBA-1 IR had weak GAL-3 IR without co-localization with IBA-1-positive cells (bottom panel). Scale bars, 5 μm.

**Table 1 vetsci-10-00403-t001:** Patient signalment, clinical characteristics, and application of archived tissue utilized in this study. Abbreviations: FI: female, intact; FS: female, spayed; IF: immunofluorescence; MC: male, castrated; MI: male intact; NCI: National Cancer Institute; IF: immunofluorescence; WB: Western blot. WHO: World Health Organization.

Group		Breed	Age (years)	Sex	Tumor Location	Tissue Analysis
NCI Grade	WHO Grade					
Low	Grade II	French Bulldog	12	FS	Right temporal lobe	IF
		Boxer	9	FS	Left piriform lobe	IF
		Boston Terrier	9	FI	Left temporoparietal lobes	IF
		Boxer	5	MC	Left frontal lobe	IF
		English Bulldog	8	MC	Left frontal to occipital lobes—periventricular	IF
High	Grade III	French Bulldog	7	FS	Right frontoparietal lobes	IF
		French Bulldog	11	FS	Left temporal lobe	IF
		Boxer	9	FS	Left cerebrum and lateral ventricle	IF
		Boxer	8	MI	Right temporoparietal lobes	IF
		Boxer	7	FS	White matter of left parietal, temporal, and occipital lobes	IF
		Boxer	13	FI	Left temporal lobe	Protein Array
		English Bulldog	12	M	Left intraventricular septum, lateral ventricle	Protein Array
		English Bulldog	9	FS	Left frontal lobe	WB, Protein Array
		Rottweiler Mix	10	MC	Left frontal lobe extending into caudate nucleus	WB, Protein Array
		Boxer	8	MI	Right piriform and temporal lobes	WB, Protein Array
		Boxer	4	MC	Lateral ventricle with broad base in caudate nucleus and periventricular white matter	WB, Protein Array
**Control Cases**		**Breed**	**Age (years)**	**Sex**	**Cause of Death**	**Tissue** **Analysis**
		French Bulldog	3	MC	Euthanasia: anti-coagulant toxicity; subdural and parenchymal spinal cord hemorrhage	IF
		Golden Retriever	10	MC	Euthanasia: primary spinal cord vasculitis	IF
		French Bulldog	3	MC	Euthanasia: ascending/descending myelomalcia following L3-L5 intervertebral disc extrusion	IF
		French Bulldog	3	MC	Euthanasia: ascending/descending myelomalcia following L3-L4 intervertebral disc extrusion	WB, Protein Array
		Beagle Mix	7	FS	Euthanasia: acute T3-L3 myelopathy; paraplegic, absent nociception.	WB, Protein Array
		Boxer	10	MC	Euthanasia: prostatic mass with urethral obstruction	WB, Protein Array
		Boxer	7	FS	Euthanasia: L4-S1 peripheral nerve sheath tumor	WB, Protein Array

## Data Availability

The raw data analyzed in this study are not available as Appendix A, though requests to the corresponding author can be made for those interested in further information.

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
