# Peer review of "Intra- and Intertumoral Microglia/Macrophage Infiltration and Their Associated Molecular Signature Is Highly Variable in Canine Oligodendroglioma: A Preliminary Evaluation"

_vetsci, 2023, doi:10.3390/vetsci10060403_

Round 1

Reviewer 1 Report

In the manuscript the authors quantified the microglial cell/macrophages by using immunofluorescence (IF) and assessed the expression of proteins, growth factors and cytokines associated with pro-tumorigenic effects in canine oligodendrogliomas. The topic is relevant and new, but the sample is small and the selection of the control cases present some flaws. Canine oligodendroglioma are not common tumors, this type of study would certainly benefit from a multi-institutional recruitment of cases. The limitation in the number of cases and controls could had been the reason for some results with borderline statistical significance (for example, the relative expression of GAL-3 in high-grade oligodendroglioma tissue compared to normal brain, p=0.060). The study should be extended by including the same number of tumor cases and controls, stratified by age.

Major points:

Objectives (lines 80-82): the sentence is vague. The authors should clearly state the specific goals of the study, including the quantification of the microglial cells/macrophages by IF and the assessment of the expression of the proteins (indicating which proteins were studied and the techniques used).

Sample size

1) The number of the cases and controls is only displayed in the Table 1. For the sake of clarification of the readers, this key information should be integrated in the text and in abstract.

2) Why only 7 cases were used as control? The number of controls should be at least the same as controls and the age of the controls should be stratified with the age of the cases. It should be hypothesized that the number of glial cells and the protein expression patterns could change with the increased age.

3) Even in these 7 control cases, why only 4 were used for the immunofluorescence and the other 3 were used for the protein expression quantification?

4) The same question applies to the tumor cases: all the low-grade cases (n=5) and only 5 high grades were used for IF. The protein expression was only analyzed in high grade tumors. What are the reasons for these methodological approaches? Considering that all the cases have biopsy or post-mortem tissue samples of the tumors, the IF could have been applied to all the cases. The IF should be performed in all the cases so more robust results would be obtained.

Table 1

The survival time was included in the table. Was this data used in this study? If not, this should not be included in the table. Moreover, if in some cases the tumoral tissue necropsy within 1 h following euthanasia, how was the survival time estimated in these cases?

Results

Did the quantification of the microglial cells by IF in normal brain been done in the same anatomical area in all the cases? The specific anatomical area studied in the control cases should be indicated.

Author Response

Reviewer 1:

In the manuscript the authors quantified the microglial cell/macrophages by using immunofluorescence (IF) and assessed the expression of proteins, growth factors and cytokines associated with pro-tumorigenic effects in canine oligodendrogliomas. The topic is relevant and new, but the sample is small, and the selection of the control cases present some flaws. Canine oligodendroglioma are not common tumors, this type of study would certainly benefit from a multi-institutional recruitment of cases. The limitation in the number of cases and controls could had been the reason for some results with borderline statistical significance (for example, the relative expression of GAL-3 in high-grade oligodendroglioma tissue compared to normal brain, p=0.060). The study should be extended by including the same number of tumor cases and controls, stratified by age.

Response: We greatly appreciate your comments and your careful review of this manuscript. Your feedback was very helpful for us in revising the manuscript. We agree with you that multi-institutional case recruitment would provide more power to a study like this. We have established collaborations for a multi-institutional biorepository with 2 additional veterinary teaching hospitals and we are optimistic that future studies will include a more robust case and control selection. However, we feel that these results are still valid as a stand alone publication without further cases added. We have previously published a similar study in canine astrocytomas with comparable case and control numbers. Even with a relatively low sample size, GAM density and expression of pro-tumorigenic proteins were more robustly and consistently increased within individual tumors and across tumors of the same grade in canine astrocytomas. Therefore, this study really highlights the potential that the GAM response is different between these two glioma subtypes. Moving forward, the field should analyze these tumors independently to futher determine similarities and differnces between oligodendroglioamas and astrocytomas.  

Concern 1: Objectives (lines 80-82): the sentence is vague. The authors should clearly state the specific goals of the study, including the quantification of the microglial cells/macrophages by IF and the assessment of the expression of the proteins (indicating which proteins were studied and the techniques used).

Response 1: We appreciate the opportunity to better articulate the scope of this project. We have adjusted the specific goals of the study to inform the reader on of our 2 major goals and techniques used. As we assayed >25 proteins, we did not include a comprehensive list, but rather stated ‘GAM-derived pro-tumorigenic molecules through canine-specific protein arrays and western blot’; Lines 95-98.

Concern 2: The number of the cases and controls is only displayed in the Table 1. For the sake of clarification of the readers, this key information should be integrated in the text and in abstract.

Response 2: We thank the reviewer for pointing out this oversight. Numbers for the different study populations have been added to the abstract (Line 40), the text (204-205, 249-250, 257-258, 261-264, 274, 286-287) and figure legends (Lines 640 and 677).

Concern 3: Why only 7 cases were used as control? The number of controls should be at least the same as controls and the age of the controls should be stratified with the age of the cases. It should be hypothesized that the number of glial cells and the protein expression patterns could change with the increased age.

Response 3: We agree with the reviewer that our control population is not as robust as we would have liked. Moreover, dog age likely has a significant impact on microglial density and phenotype, as we have previously reported in canine lumbar spinal cord. However, this study was performed with archived tissue within our personal laboratory and the Veterinary Anatomical Pathology service at UC Davis. Unfortunately, removal of the CNS is not part of routine necropsy at UC Davis; it is only removed when requested for disease diagnosis. Therefore, our archive is full of diseased brains and few normal brains. We have performed other studies utilizing normal brains and try to incorporate new cases into our control population, which really has limited our normal brain FFPE archive. We are actively addressing this critical issue.

Despite these limitations, the relative GAM density observed in this study’s control population was similar to what has been described in rat1 and canine brain2, but slightly higher than what has been described in human brain3. Moreover, protein analysis in this study was performed samples from age-matched dogs.

Concern 4: Even in these 7 control cases, why only 4 were used for the immunofluorescence and the other 3 were used for the protein expression quantification?

Response 4: As described above, tissue availability was the limiting factor. The archived fresh frozen tumor samples were from our laboratory, where it had not been protocol to collect our own FFPE. Conversely, the UC Davis Veterinary Anatomical Pathology does not collect fresh frozen samples. Thus, we did not have matched FFPE and fresh frozen samples. As mentioned above, our laboratory is actively working to correct this malalignment.

Concern 5: The same question applies to the tumor cases: all the low-grade cases (n=5) and only 5 high grades were used for IF. The protein expression was only analyzed in high grade tumors. What are the reasons for these methodological approaches? Considering that all the cases have biopsy or post-mortem tissue samples of the tumors, the IF could have been applied to all the cases. The IF should be performed in all the cases so more robust results would be obtained.

Response 5: We were unable to obtain enough fresh frozen tissue from low-grade oligodendrogliomas for analyses. The protein arrays required a minimum concentration of protein (1.5mg/mL), which we were unable to meet. As mentioned in response 4, fresh frozen tissue and FFPE tissue had been previously collected separately. Therefore, matched samples just didn’t exist. We recognize this limitation, and we are actively working to correct this malalignment.

Concern 6: The survival time was included in the table. Was this data used in this study? If not, this should not be included in the table. Moreover, if in some cases the tumoral tissue necropsy within 1 h following euthanasia, how was the survival time estimated in these cases?

Response 6: Survival time was not analyzed in this study. Thank you for pointing out that inclusion in the table is not appropriate for this manuscript. It has been removed.

Concern 7: Did the quantification of the microglial cells by IF in normal brain been done in the same anatomical area in all the cases? The specific anatomical area studied in the control cases should be indicated.

Response 7: This is a very good question and opportunity to expand on our methods in this manuscript. Microglial quantification was performed at the level of the fronto-parietal lobes in all cases. This has been added to the methods, Lines 146-147.

Reviewer 2 Report

A well written paper, a few minor points to consider:

1) L38-40 – The disease prevalence rate from the paper cited seems to say it is 0.9% for all glial neoplasms (i.e. it is not just for oligodendrogliomas); also is there a reference for the percentage given for oligos as compared to all brain tumours?

2) L132 says the protein arrays were on astrocytomas not oligodendrogliomas 

3) L342-345 - the reference at the end of the 2nd sentence could move to the end of the first sentence when you first mention the paper.

4) Throughout the paper sometimes mixed use of glioma vs oligodendroglioma e.g. 172-207 initially mentions grade II and III oligos and then moves to LGG and HGG.  which makers it unclear for the the reader if always referring to the oligodendroglioma subtype or if including other glioma subtypes .  

5)  L375-377 - which paper is this referring to?

Unfortunately the supplementary table was not available to review so I cannot comment on that. 

Author Response

Reviewer 2:

A well written paper, a few minor points to consider:

Response: We greatly appreciate your enthusiasmm for this work. Thank you for your comments and careful review of this manuscript. Your feedback was very helpful for us in revising the manuscript.

Concern 1: L38-40 – The disease prevalence rate from the paper cited seems to say it is 0.9% for all glial neoplasms (i.e. it is not just for oligodendrogliomas); also is there a reference for the percentage given for oligos as compared to all brain tumours?

Response 1: We thank the reviewer for allowing us to clarify this point. We have adjusted the text to read ‘With a disease prevalence of 0.9% for all glial neoplasms 4, oligodendrogliomas account for approximately 23% of all canine primary brain tumors 4’ (Lines 53-54).

Concern 2: L132 says the protein arrays were on astrocytomas not oligodendrogliomas 

Response 2: Thank you for catching this oversight. This has been corrected (Line 163).

Concern 3: L342-345 - the reference at the end of the 2nd sentence could move to the end of the first sentence when you first mention the paper.

Response 3: The authors are unclear which sentences the reviewer is referring to. On our original lines 342-345 encompass the following: ‘Contrary to our hypothesis, there was a paucity of GAL-3 IR in regions rich with GAM, but there was robust IR for GAL-3 co-localizing with neoplastic oligodendrocytes. Therefore, neoplastic oligodendrocytes are the likely source of GAL-3 in canine oligodendroglioma, like neoplastic astrocytes are the primary source of GAL-3 in pediatric pilocytic astrocytomas 5. However, GAL-3 is likely to effect GAM function in canine oligodendroglioma, as it is implicated in immunosuppression and resistance to immunotherapies in GBM 6.’ Review of the references seem appropriately placed. Please advise further on this point so we can make the appropriate changes.

Concern 4: Throughout the paper sometimes mixed use of glioma vs oligodendroglioma e.g. 172-207 initially mentions grade II and III oligos and then moves to LGG and HGG.  which makers it unclear for the reader if always referring to the oligodendroglioma subtype or if including other glioma subtypes.  

Response 4: Thank you for pointing out this discrepancy and allowing us the opportunity to be more consistent. We have changed nomenclature to LGG and HGG consistently throughout the manuscript text, figures, and figure legends (with the exception of Table 1).

Concern 5: L375-377 - which paper is this referring to?

Response 5: Is the review referring to this sentence: ‘Additional studies, including single cell RNA sequencing and spatial transcriptomics/proteomics, are necessary to further characterize the similarities and differences in GAM function between canine glioma subtypes to inform appropriate translational application of canine glioma clinical trials and appropriate therapeutic approaches for canine glioma patients’? This is not a reference, but the authors’ collective opinion on additional studies that are needed for further characterize the immune response across canine glioma subtypes.

(1) Kongsui, R.; Beynon, S. B.; Johnson, S. J.; Walker, F. R. Quantitative assessment of microglial morphology and density reveals remarkable consistency in the distribution and morphology of cells within the healthy prefrontal cortex of the rat. Journal of Neuroinflammation 2014, 11 (1), 182. DOI: 10.1186/s12974-014-0182-7.

(2) Toedebusch, R.; Grodzki, A. C.; Dickinson, P. J.; Woolard, K.; Vinson, N.; Sturges, B.; Snyder, J.; Li, C.-F.; Nagasaka, O.; Consales, B.; et al. Glioma-associated microglia/macrophages augment tumorigenicity in canine astrocytoma, a naturally occurring model of human glioma. Neuro-Oncology Advances 2021. DOI: 10.1093/noajnl/vdab062 (acccessed 5/17/2021).

(3) Melchior, B.; Puntambekar, S. S.; Carson, M. J. Microglia and the control of autoreactive T cell responses. Neurochemistry international 2006, 49 (2), 145-153. DOI: 10.1016/j.neuint.2006.04.002  From NLM.

(4) Song, R. B.; Vite, C. H.; Bradley, C. W.; Cross, J. R. Postmortem Evaluation of 435 Cases of Intracranial Neoplasia in Dogs and Relationship of Neoplasm with Breed, Age, and Body Weight. Journal of Veterinary Internal Medicine 2013, 27 (5), 1143-1152. DOI: 10.1111/jvim.12136.

(5) Paixao Becker, A.; de Oliveira, R. S.; Saggioro, F. P.; Neder, L.; Chimelli, L. M.; Machado, H. R. In pursuit of prognostic factors in children with pilocytic astrocytomas. Child's nervous system : ChNS : official journal of the International Society for Pediatric Neurosurgery 2010, 26 (1), 19-28. DOI: 10.1007/s00381-009-0990-8  From NLM.

(6) Fortuna-Costa, A.; Gomes, A. M.; Kozlowski, E. O.; Stelling, M. P.; Pavão, M. S. Extracellular galectin-3 in tumor progression and metastasis. Frontiers in oncology 2014, 4, 138. DOI: 10.3389/fonc.2014.00138  From NLM. Nangia-Makker, P.; Balan, V.; Raz, A. Regulation of tumor progression by extracellular galectin-3. Cancer Microenviron 2008, 1 (1), 43-51. DOI: 10.1007/s12307-008-0003-6  From NLM.

Round 2

Reviewer 1 Report

In this version the authors included relevant missing points, namely the number of samples analyzed by each technique thought the text and the  information regarding the anatomical area studied in controls.

A disagree that the fact similar sample size had been used in a previous published paper can be a valid justification for the used sample size in the present study. The recognition that the sample size is small and that more controls are needed should be included not only in the response letter, but also in the discussion of the manuscript. Limitations of the study are missing and the authors should address this point.

The abstract is still vague in the objectives of the study and did not include a clear sentence regarding the analyzed sample of tumors and controls. A revision of the abstract is necessary.

Considering the author’s reply that efforts are being made to have a large number of normal and tumoral samples and larger studies are being planned, I suggest that the title of the paper should be revise in order to clarify that this study presents preliminary evidence/results or is the first approach/assessment of this topic.

Author Response

Dear Reviewer,

We appreciate your timely and thorough evaluation of our revisions. We hope that we have addressed your concerns below.

Concern 1: A disagree that the fact similar sample size had been used in a previous published paper can be a valid justification for the used sample size in the present study. The recognition that the sample size is small and that more controls are needed should be included not only in the response letter, but also in the discussion of the manuscript. Limitations of the study are missing and the authors should address this point.

Response 1: This is a good point. We have added a limitations section at the end of the discussion for transparency, see lines 453-469.

Concern 2: The abstract is still vague in the objectives of the study and did not include a clear sentence regarding the analyzed sample of tumors and controls. A revision of the abstract is necessary.

Response 2: Thank you for the opportunity to improve clarity of the study in the abstract. We have added the following, ‘Here we quantified intratumoral GAM density of low- and high-grade oligodendrogliomas compared to normal brain, as well as the intratumoral concentration of several known GAM-derived pro-tumorigenic molecules in high-grade oligodendrogliomas compared to normal brain.’ See lines 37-40.

Concern 3: Considering the author’s reply that efforts are being made to have a large number of normal and tumoral samples and larger studies are being planned, I suggest that the title of the paper should be revise in order to clarify that this study presents preliminary evidence/results or is the first approach/assessment of this topic.

Response 3: This makes sense. We have changed the title of the paper to ‘Intra- and intertumoral microglia/macrophage infiltration and their associated molecular signature is highly variable in canine oligodendroglioma: a preliminary evaluation’.